# An Overview of Robotic Capsules for Drug Delivery to the Gastrointestinal Tract

**DOI:** 10.3390/jcm10245791

**Published:** 2021-12-10

**Authors:** Pablo Cortegoso Valdivia, Alexander R. Robertson, Nanne K. H. De Boer, Wojciech Marlicz, Anastasios Koulaouzidis

**Affiliations:** 1Gastroenterology and Endoscopy Unit, University Hospital of Parma, University of Parma, 43121 Parma, Italy; cortegosopablo@yahoo.it; 2Department of Gastroenterology, Western General Hospital, Edinburgh EH4 2XU, UK; alexanderrrobertson@hotmail.co.uk; 3Department of Gastroenterology and Hepatology, Amsterdam University Medical Centre, AGEM Research Institute, VU University, 1105 AZ Amsterdam, The Netherlands; khn.deboer@amsterdamumc.nl; 4Department of Gastroenterology, Pomeranian Medical University, 70-204 Szczecin, Poland; wojciech.marlicz@sanprobi.pl; 5Department of Public Health and Social Medicine, Pomeranian Medical University, 70-204 Szczecin, Poland; 6AJM Med-i-Caps Ltd., Nicosia 2020, Cyprus

**Keywords:** capsule, drug, delivery, technology, innovation, digestive, gastrointestinal, capsule endoscopy

## Abstract

The introduction of capsule endoscopy two decades ago marked the beginning of the “small bowel revolution”. Since then, the rapid evolution of microtechnology has allowed the development of drug delivery systems (DDS) designed to address some of the needs that are not met by standard drug delivery. To overcome the complex anatomy and physiology of the gastrointestinal (GI) tract, several DDS have been developed, including many prototypes being designed, built and eventually produced with ingenious drug-release mechanisms and anchoring systems allowing targeted therapy. This review highlights the currently available systems for drug delivery in the GI tract and discusses the needs, limitations, and future considerations of these technologies.

## 1. Introduction

The pressing need for small bowel (SB) endoscopy, with the aspiration of minimizing the discomfort of invasive gastrointestinal (GI) examination [1], has driven creative collaborations. These have disrupted the status quo of conventional GI endoscopy [2,3]. Capsule devices have been in existence for over twenty years. Although capsule endoscopy (CE) has established itself as the prime modality for investigation of the SB [4], it has taken a long time and a pandemic to emerge as a leader in other areas. A wide range of adaptations and evolutions have led to esophageal [5] and colon [6] capsules. Despite the rapid advancements in the field of artificial intelligence (AI) in CE, at present, the era of a single device for a thorough, pan-enteric investigation remains a long way away at this point. However, it seems likely that there could, in the future, be a single CE that could provide not only images but also on-the-spot diagnosis and therapy. For this to happen, a series of prerequisites, such as external capsule control, locomotion, accurate localization, and a long-lasting and reliable power supply need addressing.

The commercially available SB capsules range in size and construct between manufacturers but follow the same basic principles. They are generally between 3 and 4.5 g in weight and approximately 26 × 11 mm in size [7]. This compact casing houses a recorder or data transmitter, white-light-emitting diodes, a lens capable of magnified and high-speed photography and an internal battery. The evolution of this design has led to adaptations, such as allowing magnetic control and extended battery time. Unfortunately, these additions come at the expense of size. Colon capsule devices, for example, require a second camera head and are slightly longer, such as the PillCam™ COLON2 (Medtronic, Minneapolis, MN, USA) which is 32.3 × 11.6 mm. Improvements, such as these, have opened exciting possibilities and the prospect of using the CE devices as a vehicle for targeted therapy has been realized.

Several challenges are associated with the anatomical structure and varied physiology of the GI tract, which either prevent or complicate the dosing of specific orally given drug formulations. Medications, for example, that exhibit low solubility, low stability and/or poor permeation can have an unpredictable absorption and, therefore, unreliable bioavailability [8]. Despite several efforts to improve the pharmacokinetic drug profiles, these challenges continue to limit the field of oral drug delivery (DD). In some respects, the situation resembles the suboptimal outcomes of a carpet-bombing campaign, with too much gun powder, too little targeted damage and too many innocent casualties. It is obvious, therefore, that precision ‘bombing’ with clever ammunition loaded into intelligent (capsule) vehicles would be preferable.

The purpose of this review is to provide a wide overview of capsule-driven drug delivery systems for the gastrointestinal tract, starting from early prototypes; an in-depth focus on the currently available devices is provided.

### Gastrointestinal Environment

Currently available SB CEs pass through the entire GI tract from the mouth to the anus by autonomic peristaltic movements (passive locomotion). The upper GI tract traditionally consists of the mouth, esophagus, stomach, and proximal duodenum. The lower GI tract is divided into the (remaining) small and large intestines. Within seconds, food passes through the esophagus, but it takes approximately four to five hours for the stomach to be emptied. The colonic transit time (TT) is highly variable, but it is often one to two days. The diameter of the GI tract segments differs, with the smallest being 2.5 to 3 cm in the SB. This, of course, can be reduced dramatically in stricturing disease. The total length of the entire tract is, again, variable, but it is roughly 8–9 m and dependent on the size of the person. The mucosa of the SB (duodenum, jejunum, and ileum) is coated in villi that are crucial for absorbing food after its degradation by secreted enzymes and bile, and the large bowel primarily absorbs water. Increasingly, however, we realise that the GI tract functions are infinitely complex, and our understanding of its balanced physiology is, in many ways, basic.

CE devices with diagnostic and therapeutic purposes must cope with the changing and different environments encountered through the GI tract, such as the TT, quantity of fluids, diameter and shape [9]. Depending on the area of interest, a device may need specific adaptations and protocol designs. For example, if wishing to treat SB Crohn’s disease with the localized delivery of drugs, one may need a CE device that has a relatively small diameter and has an efficient modality for prolonged drug release.

## 2. Direct Drug Delivery Systems

Although drug delivery systems (DDS) have been developed for use in other target areas [10], the complex anatomy and physiology of the GI tract come with several issues that need to be addressed. These include a releasing mechanism for the drug and an anchoring system to provide a stable position in the desired area. Therefore, several micromechatronic systems have been developed. The statuses of such capsules fall into the categories of prototype, patent, concept and commercial. This overview will mainly focus on the commercial group.

### 2.1. Releasing Mechanisms

DDS differ in their releasing mechanisms, which can be passive or active. In the former, the chemical is exposed to the GI environment whenever an external trigger (e.g., pH or temperature). In the latter, the active expulsion of the drug is driven by the remote activation of a release mechanism. The main advantage of active mechanisms is enhanced control over DD.

#### 2.1.1. Passive Release Mechanisms

The first example in this field is the high-frequency (HF) capsule [11], developed in Germany in the 1980s. In this system, the drug release is triggered by an external radiofrequency generator that heats and melts a thread, thus piercing a latex balloon containing the chemical with a released needle [12].

A similar mechanism is also exploited by InteliSite^®^ (Scintipharma Inc., Lexington, KY, USA). The heat generated by an external radiofrequency signal activates a spring mechanism in the inner cage of the capsule, which pushes the content of the reservoir through the outer shell (capacity 0.8 mL) [13].

Another example of a passive mechanism is the magnetic active agent release system (MAARS) (Mathesy GmbH, Jena, Germany). The carrier capsule consists of two magnetic parts bond together and contained a reservoir (liquid or powder of up to 0.847 mL). The drug release is performed by an external magnetic field that separates the capsule's two components [14].

More recently, another releasing system was explored by the RaniPill™ (Rani Therapeutics, San Jose, CA, USA). A hydroxypropyl methylcellulose capsule enclosed in an enteric coating, preventing its dissolution in the low pH of gastric acid, contains a robotic autoinjector and a self-inflating balloon. When the pill reaches the SB (higher pH), the coating dissolves, inflating the balloon. The balloon inflation exposes a dissolvable needle, allowing drug delivery via injection through the intestinal wall [15]. This system was initially developed to be used in chronically ill patients requiring repeated subcutaneous or intramuscular injections. In addition, these systems have been suggested for medications such as octreotide, TNFα inhibitors, parathormone or insulin.

Although appealing, these systems come with limitations. For example, the activation may fail due to interposed tissues weakening the electromagnetic signal; the content may leak if the sealing of the outer shell isn’t perfect; and, the site targeting before the release lacks precision and practicality (e.g., the InteliSite^®^ capsule relies on gamma scintigraphy). To overcome these limitations, active systems were subsequently developed.

#### 2.1.2. Active Release Mechanisms

The single-use electronic IntelliCap^®^ device from Medimetrics (Eindhoven, The Netherlands) consists of a cap (containing 0.3 mL of liquid drug), microprocessor, battery, pH and temperature sensors, RF transceiver and pump [16]. Drug release can be manually controlled through a laptop or pre-programmed to release at a particular point within the GI tract. The device monitors resident times throughout the GI tract and local temperatures and pH with wireless data transmission. This allows automated drug release with pH/temperature-based positioning, which correlates well with scintigraphy [17]. In addition, the screw-pump release system can be used for fluids, pastes or suspensions in a tailored fashion depending on the therapeutic or research intention [18,19]. 

In 2009, Groening and Bensmann presented a capsule-shaped model with 28 × 8.5 mm dimensions that contained a gas-producing cell and a receiver system [20]. The body of this capsule is manufactured using a polypropylene cylinder (14.5 × 6.7 mm) including a 4.8 mm long piston separating the gas-producing cell and the drug reservoir. The drug reservoir contains 0.17 mL of an aqueous solution of oxprenolol hydrochloride (0.7 g/mL) and the solution is released through a 0.6 mm borehole in the rubber form sealing. The gas-producing cell (3.5 × 7.8 mm) (size 3, Simatec, Herzogenbuchsee, Switzerland) is glued onto the cylinder. The electric circuit is placed next to the gas-producing cell, consisting of a small coil, an SMD trim condenser, two SMD Skottky diodes and a MOSFET transistor. Therefore, only 16% of the capsule volume is used for the loaded drug, and this device has a 1 h activation time for releasing the dose.

Magnetic activation mechanisms are also exploited in the magnetically actuated soft capsule endoscope (MASCE). This consists of two permanent magnets that, when activated, compress a soft capsule, thus allowing the release of the drug from a 0.17 mL internal chamber [21]. The main advantage of this system is the possibility of delivering multiple doses of the drug at a pre-established rate.

A magnetic-field-driven mechanism is also used by the Enterion™ capsule (Phaeton Research, Nottingham, UK) (Figure 1). This 32 × 11 mm capsule contains a reservoir with a 1 mL capacity and is equipped with a spring mechanism which can be activated via a magnetic impulse. An internal heating element triggers the capsule’s opening and rapidly ejects the drug with a piston [22]. This system has similar features to the InteliSite^®^ capsule, with a comparison shown in Table 1.

In 2010, Pi et al. [23] proposed a solid propeller micro-thruster for the actuation assembly of a patented remotely controlled capsule (RCC) (Figure 2) [24]. It produces sufficient gas pressure to empty the drug reservoir using a microigniter as the critical component of the miniaturized thruster that houses diazodinitrophenol as a detonating agent and black powder as a propellant. Results demonstrated that 166 mW of power consumption led to successful combustion with a complete and rapid drug release achieved when the propellant ranged from 16 mg to 20 mg. This system could provide a promising alternative for site-specific DD in the human GI tract. A problem with these two models is that they are devoid of imaging guidance.

In 2013 Woods and Constandinou [25] presented their concept design for a microrobot platform with a max volume of 3.0 cm^3^, equipped with pH, temperature and pressure sensors and a complementary metal-oxide-semiconductor (CMOS) for real-time image guidance. This design contains two sub-systems: a micro-positioning mechanism that can deliver 1 mL of targeted medication and an anchoring mechanism whose function is to hold the capsule in place against GI peristalsis. Of course, the overall volume and geometry of the system is greatly governed by the constraints imposed by the size that can be swallowed. Pivotal work in this area showed that a volume of 3.0 cm^3^ can be swallowed [26]. The platform is operator dependent as one caregiver must go through the data in real-time to identify a defined target site and remotely deploy the anchoring mechanism.

Furthermore, the operator controls the rotation and advancement of the needle for the localized medication infusion (Figure 3). The needle can be placed in a 360° range while simultaneously maintaining a diametrically opposite relationship with the holding mechanism. This feature ensures penetration of the GI tract wall. The authors envisaged that the system could be used to detect and treat GI tract pathologies such as SB Crohn’s disease or tumors.

In 2016, Le and colleagues presented a DD prototype known as the Active Locomotive Intestinal Capsule Endoscope (ALICE) [27]. This prototype had been introduced by the same group a year earlier. This device consists of two soft ring-type magnets and a simple plastic hinge [28]. The ring-type magnets are axially magnetized, attracting each other and keeping the drug enclosed inside the module. The ALICE system provides controlled navigation for the integrated ALICE and DD module to investigate and accurately infuse the drug to the lesion area. First, the rings are demagnetized, and a strong pulsating magnetic field is applied in a radial direction, causing the enclosed drug to be released. Then, the two rings are axially magnetized and attract each other again, and thanks to the plastic hinge, the DD module is returned to its initial shape. The integrated DD module of the ALICE system has a diameter of 12 mm and a length of 33 mm and has been tested in vitro. 

In 2017, the SonoPill Programme team presented their concept of a prototype device named SonoCAIT (Figure 4). This capsule, with the dimensions 10 × 30 mm, has four main components necessary for ultrasound-mediated targeted DD: a focused US transducer, a DD channel, a video camera and a light source [29], with the detailed fabrication process available elsewhere [30]. Initially tethered to facilitate delivery power and reagents, it was suggested that future versions will include wireless communications and power delivery. This could be achieved by embedding an antenna in the capsule shell to maximize the space within the capsule. The focused US source (4 MHz frequency) consisted of a self-focusing bowl with an outer diameter of 5 mm. The DD channel consisted of fine-bore polythene tubing with an outer diameter of 0.96 mm and an inner diameter of 0.58 mm. Video imaging is provided by a CMOS camera, giving a resolution of 220 × 224 pixels. Illumination is provided by a printed circuit board (PCB) with four 40 mW LEDs (OSRAM Opto Semiconductors GmbH, Germany). The PCB has a 1.5 mm diameter hole in the center to allow the camera to pass through.

### 2.2. Anchoring Systems

To target an area of interest within the intestine, resisting peristalsis is necessary to “anchor” in the area. Various mechanisms have been proposed for this. The most basic of these being string attachments which allow the capsule to be held in the esophagus, giving increased views by preventing rapid transit and allowing the capsule to be retrieved [31,32]. Alternatives trialed for anchoring within the lumen include balloon-shaped, magnetic and “leg”-based designs to hold the capsule in position. These provide an increasing anchoring force with a wider diameter to the clamping component and textured contact surfaces were shown to give higher friction but had no effect from the robot weight [33].

Inspired by insect feet, legged devices have been in development since 2007 with the aim of locomotion within the lumen. A 12-legged capsule device designed for locomotion within the colon moves through the rotation of two sets of legs [34]. A three-legged device with micropillar adhesive feet was designed in 2008, which holds onto the intestinal wall and can withstand the forces of peristalsis [35]. Legged devices have been shown to have increased anchoring power with the addition of an external magnetic field [36].

A technical specification overview of the discussed DDS is summarized in Table 2.

## 3. Conclusions

With shrinking technologies and microprocessors, the 3.0 cm^3^ of traditional available “real estate” within a capsule endoscope is now able to pack far more in. Although used routinely for diagnostics, this increasingly allows adaptations and extra features, with magnetic control, prolonged battery life, second camera heads, etc. already available. The direct visualization and real-time monitoring offered, along with the extra storage capacity, are now beginning to allow drug, or non-drug therapy, to be delivered. This therapy is being targeted in a personalized, remotely controlled and precise fashion within the GI tract.

Robotic capsules are getting ready to hit prime time. The use of capsules only as diagnostic imaging devices will soon be a thing of the past. Just like the platform′s path in the arena of digestive diagnostics was dependent on the maturity of the required technologies, its path in drug delivery has taken longer, due to the readiness level of the required integrated modules. For instance, one practical application of sensing/delivering robots could be the treatment of a localized GI tract pathology by releasing hemostatic powder or a similar spray formulation and potent anti-inflammatory agents in points of interest without the need for an uncomfortable intervention or systemic side effects. Another useful application will be the delivery of systemically required medications, e.g., insulin or chemotherapeutic agents. Moreover, a combination of both local and systemic drug delivery by robotic capsules would be of interest in treating autoimmune-mediated bowel diseases.

## Figures and Tables

**Figure 1 jcm-10-05791-f001:**
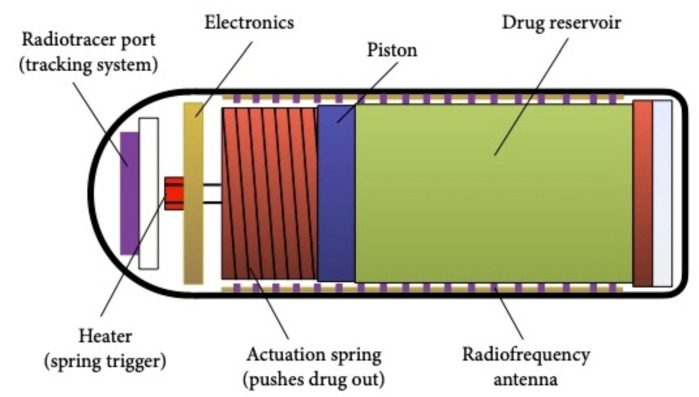
Schematic drawing of the Enterion™ capsule.

**Figure 2 jcm-10-05791-f002:**
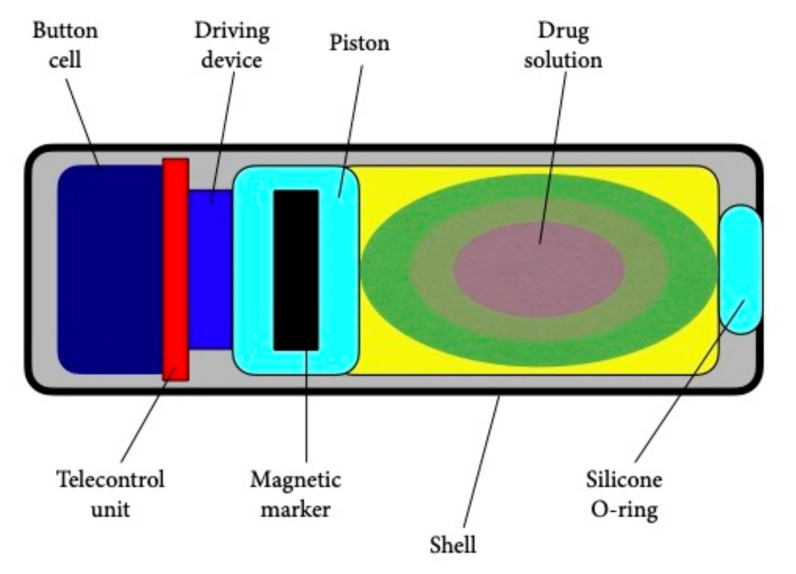
Schematic drawing of the remotely controlled capsule (RCC).

**Figure 3 jcm-10-05791-f003:**
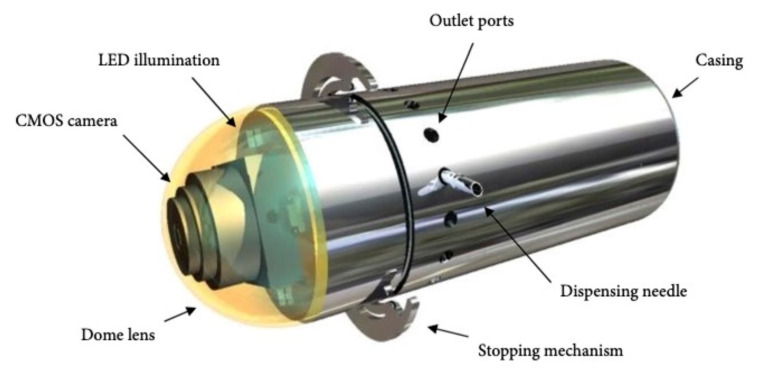
A prototype capsule with an integrated anchoring mechanism and a targeted medication delivery system [with permission from Timothy G. Constandinou].

**Figure 4 jcm-10-05791-f004:**
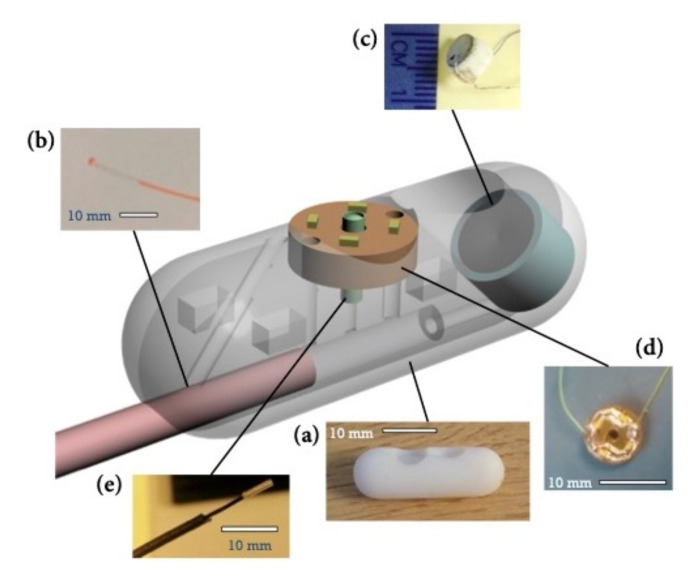
A prototype capsule for ultrasound-mediated targeted drug delivery, (**a**) the capsule body, (**b**) the tether with the electrical cables and the drug delivery channel, (**c**) the ultrasound source, (**d**) the light source and (**e**) the miniature camera [with permission from Sandy Cochran].

**Table 1 jcm-10-05791-t001:** Comparison between the InteliSite^®^ and the Enterion™ drug delivery systems.

	InteliSite^®^ Capsule	Enterion™ Capsule
**Seal**	Thin layers of lubricant between a two-sleeve system prevents drug leakage	Leakage from the drug reservoir is avoided by a compressed silicone ring seal
**Activation**	Activation energy is transmitted from the outside. Activation can take up to 2 min	Activation energy is inside the capsule. The energy is released via a radio-frequency transmitter
**Expulsion**	Expulsion is passive and slow	Expulsion is active and fast via a spring-powered piston
**Feedback mechanisms**	Absent	Present
**Types of drugs**	Solutions, low-viscosity formulations.	Wide range of formulations

**Table 2 jcm-10-05791-t002:** Technical specifications or drug delivery systems.

DDS.	Dimensions (mm)	Reservoir Volume (mL)	Release Mechanism
**HF**	-	-	Passive
**InteliSite^®^**	35 × 10	0.8	Passive
**MAARS**	18.2 × 7.7	0.34	Passive
**RaniPill™**	26.1 × 10	-	Passive
**IntelliCap^®^**	27 × 11	0.3	Active
**Groening prototype**	28 × 8.5	0.17	Active
**MASCE**	40 × 15	0.17	Active
**Enterion™**	32 × 11	1	Active
**RCC**	30 × 10	0.7	Active
**Woods prototype**	36 × 11.1	1	Active
**ALICE**	33 × 12	0.78	Active
**SonoCAIT**	30 × 10	NA	Active

Abbreviations: DDS, drug delivery system; NA, not applicable.

## Data Availability

Not applicable.

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
