# Peer review of "An Overview of Robotic Capsules for Drug Delivery to the Gastrointestinal Tract"

_jcm, 2021, doi:10.3390/jcm10245791_

Round 1

Reviewer 1 Report

This review of drug delivery system using capsule presents the tatus of currently available capsules for drug delivery in the GI tract with limitations, and future considerations.

Author Response

REVIEWER #1

This review of drug delivery system using capsule presents the status of currently available capsules for drug delivery in the GI tract with limitations, and future considerations.

Answer

Thank you very much for your kind comments.

Reviewer 2 Report

It is an interesting article that presents the drug delivery system of capsule endoscopy. It is a unique review of this field of study. However, there are some concerns about publishing this article. 

  1. The introduction generally does not include a table. The table is not necessary.
  2. The title is not suitable for this study. It should include CE. Like "An overview of the field of drug delivery systems for the gastrointestinal tract with capsule endoscopy" is preferable.
  3. In the introduction, the authors should clearly state the primary outcome and secondary outcome. In addition, they should present how to search the references.
  4. The authors should present possible adverse events precisely.
  5. The authors should present why the drug delivery system of CE is not in clinical use at this moment and how to overcome it.
  6. The number of references is small and old for the review article. However, it is permissible since there has been no new advantage in this field of study in recent years; how do you think of that?
  7. I was interested read your article.

Author Response

REVIEWER #2

It is an interesting article that presents the drug delivery system of capsule endoscopy. It is a unique review of this field of study. However, there are some concerns about publishing this article. 

  • The introduction generally does not include a table. The table is not necessary.
  • The title is not suitable for this study. It should include CE. Like "An overview of the field of drug delivery systems for the gastrointestinal tract with capsule endoscopy" is preferable.
  • In the introduction, the authors should clearly state the primary outcome and secondary outcome. In addition, they should present how to search the references.
  • The authors should present possible adverse events precisely.
  • The authors should present why the drug delivery system of CE is not in clinical use at this moment and how to overcome it.
  • The number of references is small and old for the review article. However, it is permissible since there has been no new advantage in this field of study in recent years; how do you think of that?
  • I was interested read your article.

Answer

Thank you for your suggestions: the manuscript has been edited accordingly.

1- Table 1 has been removed, subsequent tables have been re-numbered.

2- The title has been changed in “An overview of ROBOTIC CAPSULES for drug delivery to the gastrointestinal tract”. We would prefer not to include the word “endoscopy” as not all the described systems strictly fall into this definition.

3- The introduction now includes a specific sentence regarding the outcome of the study. About references, as this study was -from the beginning- intended as an overview, the reference search was not performed systematically.

4- Thank you for this comment. Nevertheless, as many of the described systems were only presented as prototypes with specific focus on the actual mechanical mechanism, adverse events were never described. In addition, due to the paucity of performed procedures (whenever compared to capsule endoscopy), it is also possible that adverse events never actually occurred. We hope this will not influence the overall impact of your precious evaluation.

5- In this paper, we performed an overview of drug delivery systems for the GI tract with capsule, starting from the early prototypes to new available systems. Some of them are actually on the market, as pointed out in the text. We decided to focus our attention on these systems, as shown in specific tables (i.e., InteliSite vs Enterion)

6- To be honest, several studies and congress papers have been published in the latest years on this topic…nevertheless, this was almost exclusively on specialized mechanical engineering papers. As the main drive of this paper was to provide an exhaustive overview on these systems especially for clinicians, we decided not to focus our attention on these exquisitely technical references.

7-On behalf of all Authors, we thank you for your valuable comments

Reviewer 3 Report

All the points raised in the first version of the manuscript have been properly addressed by the authors.

Author Response

REVIEWER #3

All the points raised in the first version of the manuscript have been properly addressed by the authors.

Answer

Thank you for your valuable time in reviewing our article.